# Effect of Cold Atmospheric Plasma (CAP) on Osteogenic Differentiation Potential of Human Osteoblasts

**DOI:** 10.3390/ijms23052503

**Published:** 2022-02-24

**Authors:** Benedikt Eggers, Anna-Maria Wagenheim, Susanne Jung, Johannes Kleinheinz, Marjan Nokhbehsaim, Franz-Josef Kramer, Sonja Sielker

**Affiliations:** 1Department of Oral, Maxillofacial and Plastic Surgery, University Hospital Bonn, 53111 Bonn, Germany; franz-josef.kramer@ukbonn.de; 2Research Unit Vascular Biology of Oral Structures (VABOS), Department of Cranio-Maxillofacial Surgery, University Hospital Muenster, 48149 Münster, Germany; a_wagenheim@icloud.com (A.-M.W.); susanne.jung@ukmuenster.de (S.J.); johannes.kleinheinz@ukmuenster.de (J.K.); 3Section of Experimental Dento-Maxillo-Facial Medicine, University Hospital Bonn, 53111 Bonn, Germany; m.saim@uni-bonn.de

**Keywords:** cold atmospheric plasma, osteoblast, osteogenic differentiation, tissue regeneration

## Abstract

Bone regeneration after oral and maxillofacial surgery is a long-term process, which involves various mechanisms. Recently, cold atmospheric plasma (CAP) has become known to accelerate wound healing and have an antimicrobial effect. Since the use of CAP in dentistry is not yet established, the aim of the present study was to investigate the effect of CAP on human calvaria osteoblasts (HCO). HCO were treated with CAP for different durations of time and distances to the cells. Cell proliferation was determined by MTT assay and cell toxicity by LDH assay. Additionally, RT-qPCR was used to investigate effects on osteogenic markers, such as alkaline phosphatase (*ALP*), bone morphogenic protein (*BMP*)2, collagen (*COL*)1A1, osteonectin (*SPARC*), osteoprotegerin (*OPG*), osterix (*OSX*), receptor activator of NF-κB (*RANK*), RANK Ligand (*RANKL*), and Runt-related transcription factor (*RUNX*)2. There were small differences in cell proliferation and LDH release regarding treatment duration and distance to the cells. However, an increase in the expression of *RANK* and *RANKL* was observed at longer treatment times. Additionally, CAP caused a significant increase in mRNA expression of genes relevant to osteogenesis. In conclusion, CAP has a stimulating effect on osteoblasts and may thus represent a potential therapeutic approach in the regeneration of hard tissue defects.

## 1. Introduction

After oral and maxillofacial surgery, the healing of hard tissue is a much more complex and long-term process as compared to soft tissue. The regeneration of bone is particularly extensive and involves many different mechanisms, such as the differentiation of osteoblasts or the formation of collagen fibres and its arrangement according to the mechanical forces [1,2,3]. The process is conducted by genes such as bone morphogenic protein 2 (*BMP2*), Runt-related transcription factor 2 (*RUNX2*), osterix (*OSX*), and osteoprotegerin (*OPG*), and followed by markers that appeared later in osteogenic and mineralization cascade, as alkaline phosphatase (*ALP*), osteonectin (*SPARC*), and collagen 1 (*COL1A1*) [4,5,6,7,8,9,10,11]. Additionally, bone is highly vascularised and permanently remodels itself in human life [12,13]. However, systemic factors such as increased age, malnutrition, or metabolic syndrome can also negatively influence bone regeneration [14,15,16].

Recently, cold atmospheric pressure plasma (CAP) has become known to improve wound healing [17,18]. CAP is a reactive gas state that can be generated in many ways, such as by argon or the ambient air [19,20]. Different technologies are known for medical applications, such as dielectric barrier discharge (DBD) devices, plasma jets, or hybrid devices [21].

Although the detailed effects of CAP have not yet been fully elucidated, it seems to cause a highly regulated wound healing–promoting mechanism in various cells and tissues. Thus, among other aspects, stimulating effects on, e.g., keratinocytes, fibroblasts, or endothelial cells have been observed, in particular an upregulation of certain genes and proteins, an increase in cell proliferation, and higher cell viability [22,23,24]. Furthermore, several studies prove the effectiveness of cold plasma, especially in dermatology [17,18]. Additionally, apoptotic effects of CAP have also been described for certain types of cancer cells [25,26,27,28].

Oral and maxillofacial surgery in particular could benefit from clinical use of CAP with the possibility of increased bone regeneration or the application of the anticancer effect. Promising studies with preosteogenic mice cell lines and human periodontal ligament cells showed an enhanced osteogenic differentiation potential by CAP [29,30,31]. In this study we wanted to observe a potential pro-proliferative effect of CAP on hard tissue cells. This effect can be used, for example, following trauma or regenerative surgery after treating a carcinoma. Several osteogenic-related markers were to be investigated to understand the effect of CAP on the factors relevant for bone growth and mineralisation. Since possible clinical use requires precise understanding of its effect on the hard tissue cells of the skull and, of course, must exclude damaging effects, the aim of the present study is to investigate the effects of CAP on human calvaria osteoblasts (HCO).

## 2. Results

### 2.1. Evaluation of the Influence of Treatment Parameters

First, we investigated the effect of different treatment lengths and effects of remaining culturing medium during treatment on the cells. As shown in Figure 1a, there is no significant difference in living/dead staining between the treatment length of 60, 90, and 120 s with a distance of 2.0 cm. We could observe slight effects in proliferation (Figure 1b) and LDH release (Figure 1c). Proliferation decreased with length. Effect induced by remaining culturing medium during treatment showed clearest results. Release of LDH is constantly higher, when medium remains on cells, particularly during the first 24 h. Based on these findings, we focused on a treatment time of 60 s for the following procedure.

When an application is performed on patients, the continuous movement of the patient and the doctor causes the distance to change continuously. As shown in Figure 1d, proliferation was not influenced by distance of the device to the cells. In addition, the unaltered LDH secretion demonstrated that CAP did not have a cytotoxic effect on the cells (Figure 1e). To summarize, distances of 1.0 cm caused the strongest negative effects on the cells, so we at first focused on distances of 0.5 and 0.1 cm.

### 2.2. Investigation of Treatment Modality

Next, we examined the effect of a single 60 s treatment versus multiple 60 s treatments distributed over several days, using an MTT assay. The CAP treatment resulted in an increase in cell viability only after about 10 d, both at a distance of 0.1 cm and at a distance of 0.5 cm. There was a clear difference between single and multiple treatments, especially after 10 and 14 d. Three treatments seem to have a negative effect on proliferation. Overall, cell proliferation was less pronounced at a greater distance between the CAP device and the cells (Figure 2a,b).

The focus of this study was the influence of CAP onto osteogenic differentiation potential. To ensure that CAP has no effects on the contrary osteoclastic differentiation potential, however, expression of osteoclast-associates markers, *RANK* and *RANKL*, were analysed additionally (Figure 3a–d).

Especially at a distance of 0.5 cm, multiple treatment causes an increase of *RANK* and *RANKL* (Figure 3b,d). Especially after 10 days, the mRNA expression of both markers is strongly increased. Interestingly, the activation of *RANK* and *RANKL* is less pronounced at a closer distance of 0.1 cm. *OPG*, a decoy receptor of *RANKL* and an osteoclastogenic inhibition factor, shows the strongest expression after 10 days (Figure 3e,f). In general, however, it appears that the upregulation of *OPG* mRNA expression is delayed with a 0.5 cm treatment compared to a 0.1 cm treatment (Figure 3e,f). We observed an upregulation of *OPG* mRNA at distances 0.1 and 0.5 cm, especially at 10 d. For 0.5 cm distance, *OPG* shows a similar expression pattern to *RANKL*. For a distance of 0.1 cm a different expression pattern to *RANKL* is visible. Only at day 4 similarities can be observed. At day 10, expression of *OPG* increased strongly in contrast to the expression of *RANK* and *RANKL*.

### 2.3. Upregulation of Osteogenic Markers

For the following experiments, we focused on a single CAP treatment, as the triple application of CAP had more damaging effects on the cells. Figure 4 summarised gene expression results of different early osteogenic markers and early mineralisation inducing marker.

Interestingly, CAP had a slight influence on the regulation of *BMP2* mRNA. However, early osteogenic markers such as *BMP2*, *RUNX2*, and *OSX* are activated slightly faster by CAP at a distance of 0.1 cm than at a distance of 0.5 cm (Figure 4a–c). However, this effect equalises over time and then the larger distance has a greater effect at 14 days. Overall, *OSX* regulation peaks at 4 d, whereas *RUNX2* peaks at 10 d for 0.1 cm distance and at 14 days for 0.5 cm distance. Similarly, for mRNA regulation of *ALP*, the lower distance of the DBD causes a stronger effect than the higher distance, with the peak at 14 days (Figure 4d). However, for *COL1A1*, CAP results in an overall downregulation of mRNA levels at 4 and 10 days, with this being more pronounced at a distance of 0.1 cm from the cells. Only at a distance of 0.5 cm at 14 days is there a slight upregulation of the *COL1A1* mRNA level. Finally, CAP induces a slight upregulation of mRNA expression in *SPARC*, especially after 10 and 14 days. Similar effects were observed for *ALP* and on late osteogenic markers *COL1A1* and *SPARC* (Figure 4d–f).

## 3. Discussion

In the present study, we have demonstrated the mineralization-promoting effect and the stimulation of osteoblastic markers by CAP on human osteoblasts. We also found that multiple treatments also have a negative effect on the cells.

In previously published studies, we have observed the influence of CAP on periodontal ligament cells and cementoblasts [23,32]. Since not only soft tissue cells play an important role in dental regeneration processes, we also wanted to investigate the effect of CAP on osteoblasts in this study.

At first, we wanted to determine the optimal parameter for CAP treatment, such as length of treatment or distance between the CAP device and the cells. Since our device is a DBD device, for which the manufacturer recommends a close distance to the tissue, but does not further specify, optimal parameters to osteoblast cell monolayer had to be investigated. Interestingly, there are few differences in viability, proliferation, and cytotoxicity at a distance of 0.1 and 0.5 cm (Figure 1d,e). Larger distances showed cytotoxic effects by higher secretion rate of LDH (Figure 1c). Referring to the literature, Takahashi et al. used an experimental DBD for irradiating human gingival fibroblast monolayers for 30 s at a distance of 1 cm and detected no difference in LDH release when compared to the control [33]. Steinbeck et al. observed a frequency-dependent increase in LDH release after 10 s DBD treatment of osteoblasts at a distance of 2 mm [34]. A distance-related increase in LDH secretion of cell monolayers has not been described for DBD plasma devices, which could be due to the fact that CAP has various components that have a different effect on cells, such as the production of reactive species or the emission of electromagnetic fields or ultraviolet radiation [35], which could have different effects at a greater distance. For example, other authors have described the effects of electromagnetic fields or UV light on LDH release [36,37]. It may be that at greater distances, only single components of the plasma affect the cells negatively. Furthermore, increased LDH secretion could also be caused by the formation of streamers, which have been observed at greater distances from DBDs [38,39]. Since the intensity of streamers is very high, cells may be negatively affected [40]. However, as the exact effects of CAP on cells and tissues are not yet fully understood, further research is needed to further elucidate the distance-related effects.

Additionally, CAP was found to have a proliferative effect on osteoblasts, as it has been shown in other studies for other cell types, such as gingival fibroblasts, MG63 cells, or cementoblasts [23,32,33,41]. Interestingly, this effect was first observed after 10 days. In contrast, fibroblasts revealed a significant increase in proliferation already after 72 h and cementoblasts and MG63 cells already after 24 h following DBD treatment [32,33,41]. It appears that the proliferative effect in osteoblasts after stimulation with CAP only becomes apparent after a longer time than in the other cells. However, possible differences could also be due to the way CAP is applied; in our experiments, the cell culture medium was removed before applying CAP to show the direct effect on the cells, since in preliminary experiments we observed negative results with remaining medium during CAP treatment (Figure 1c). Steinbeck et al. also removed culture media before the treatment and replaced it immediately after DBD treatment, although the authors did not examine cell proliferation but the number of dead cells [34]. The authors show a slight increase in dead cells due to CAP treatment at different intensity settings of the DBD, with the cells only being treated for 10 s. Since we did not observe any negative effects after 60 s, the difference could be due to the configuration of the plasma devices; Steinbeck et al. used an experimental DBD and we used a certified medical device. However, further research is necessary to understand these effects in more detail. In many other studies found in the literature, the cell culture medium is left in place [32,41,42,43]. It is known that fluids such as cell media can also be treated with CAP, both with plasma jets and DBDs, and have an attenuated biological effect on cells and tissues [44,45]. The overall CAP effect on cells covered with cell culture medium may be the result of simultaneous direct and indirect treatment. However, in our study, we wanted to focus on a direct treatment.

Additionally, we investigated that an increase of the treatment distance and multiple treatments resulted in a decrease of the proliferation effect. Apoptotic effects are well known for longer treatments with CAP; in an in vitro study, for example, treatment with DBDs longer than 40 s results in significant decrease in cell number of HaCaT keratinocytes [43]. These apoptotic effects are especially observed for neoplastic cells using DBDs and plasma jet technology [26,42,43,46]. These selective effects on benign cells, such as fibroblasts, compared to cancer cells have also been described for indirect DBD treatment [47]. However, since the authors activated the media for 20 min and we only performed an indirect treatment of 30–90 s, this may explain the weak effect of the indirect treatment on the cell monolayers in the preliminary experiments.

However, we can conclude for our device that a direct treatment of 60 s is beneficial and multiple treatments are harmful for HCO. Further detailed studies are necessary to find out how the cells may react to treatment times with CAP between 60 and 180 s.

We also analysed expression of osteoclast-associated markers. Usually, in studies investigating osteoblast expression patterns, mainly alteration in expression of well-known osteogenic markers is analysed. Possible opposite effects are unintended. In addition to a reduced proliferation, we discovered that multiple CAP applications upregulated the mRNA of *RANK* and *RANKL* more than a single application (Figure 3). These markers are crucial for bone degradation: *RANK* is the signalling receptor important for osteoclastogenesis [48] and *RANKL* its ligand [49,50].

Since we wanted to avoid damaging the bone, we focused on a one-time 60 s CAP treatment. Other authors have also described that CAP decreases *RANKL* mRNA regulation: The authors show a reduction of *RANKL* and an increase of *OPG* in periodontitis in an animal model after a 2 min treatment with an argon plasma jet [51]. However, in our experiments, we did not test the effect of a 2 min treatment with DBD CAP, but only that of a 3 × 1 min treatment. It is also possible, therefore, that the different method of CAP application is a factor, which affects *RANKL* regulation. Further studies with different CAP devices are necessary to decipher this background in more detail.

In addition to *RANKL*, regulation of *OPG* has been studied. *OPG* acts as a decoy receptor for *RANKL* and reduces bone resorption by inhibiting osteoclast differentiation and activation and stimulating osteoclast apoptosis [52]. Due to the fact that *OPG* is a decoy receptor for *RANKL*, we could observe a nearly similar expression pattern of *OPG* and *RANKL*. This could be interpreted that *OPG* estruses *RANKL*. For 0.1 cm distance, expression pattern of *OPG* and *RANKL* are similar at day 4 and at day 10 expression of *OPG* increased strongly. *OPG* does not only inhibit osteoclastogenesis, it is also an early marker for osteogenic differentiation [53].

Interestingly, we observed an upregulation of *OPG* mRNA at distances of 0.1 and 0.5 cm, especially at 10 d. The distance to the cells does not seem to have a particularly large influence on the expression of this marker, but there is a slight weaker regulation at 0.5 cm.

Further, we analysed the expression of main osteogenic relevant marker with regard to early marker and early mineralisation inducing markers. Surprisingly, *BMP2*, one of the potential inducers of osteogenic differentiation [11], has not been affected by CAP. Less is known about CAP and expression of *BMP2*. Studies focusing on the expression rate of osteogenic markers have shown a relatively low expression level of *BMP2* after both plasma jet and DBD CAP treatment [30,34]. In contrast, however, the reactive oxygen species that contribute to the CAP effect [35] have been described to control the expression of signaling molecules such as *BMP2* to mediate osteoblast differentiation [54]. However, since the exact effect of CAP is not yet understood, it is possible that the sum of all components has a different effect than the isolated effect of ROS/RNS, electric fields, or UV light. For this reason, additional research is needed to further understand the CAP effect on osteogenic differentiation.

Additionally, we focused on *RUNX2*, a key marker which is essential for the control of bone formation and the regulation of bone cell growth and differentiation [9,10]. After CAP application at a small distance (0.1 cm) an upregulation of *RUNX2* mRNA was observed at 10 d. Interestingly, for 0.5 cm distance the mRNA upregulation was only visible after 14 d. Thus, it appears that increasing the distance of the CAP device causes a delayed cell response. Similar effects regarding the stimulation of *RUNX2* have been observed in cementoblasts after 60 s of CAP application at 1 day [32]. Nevertheless, the cell response seems to take significantly longer in HCO cells than in cementoblasts. These differences may be caused by the fact that murine cementoblasts were used and treated indirectly (cells with remaining culturing medium). Further studies on murine osteoblasts or human cementoblasts could help to decipher these questions.

Further on, we observed an upregulation of *OSX*, an essential mineralization marker, which controls osteoblastic bone formation and acts downstream of *RUNX2* [55,56]. Again, stimulation at a small distance leads to early upregulation of mRNA regulation, whereas stimulation at a larger distance leads to upregulation later on, after 14 d. Upregulation of *OSX* mRNA by CAP has been shown in cementoblasts at 1 d after CAP stimulation of 60 s [32]. The reason why mouse cementoblasts react significantly faster to stimulation with CAP could be due to the different cell type. Additionally, it is well known that the mineralisation of bone takes a certain time [57]. However, the exact influence of CAP functioning of CAP is not yet fully understood.

Interestingly, CAP downregulated *COL1A1* mRNA expression at 4 and 10 days. At 14 d, a slight upregulation at a distance of 0.5 cm could be observed. *COL1A1* mRNA has been shown to be increased in the proliferation phase of bone mineralization in the first 10–12 days and is associated with the formation of the extracellular matrix. [57]. Since an immediate upregulation of *COL1A1* mRNA after CAP treatment has been shown in cementoblasts, PDL cells, and MG63 cells, HCO cells seem to react differently to CAP treatment [23,32,41].

Finally, we discovered a change in regulation of *SPARC*. *SPARC* is a key regulator of osteoblast mineralisation [58,59] and the upregulation of *SPARC* mRNA after CAP treatment was observed at both 10 and 14 d.

Regarding mineralization, expression rate of *ALP*, which is also a key factor of bone mineralization [60], was upregulated by CAP. Other authors have also investigated the effect of CAP on the mineralization of hard tissues: Interestingly, CAP treatment of murine pre-osteoblasts covered with cell-culture medium using a plasma jet resulted in higher proliferation, ALP activity, and upregulation of osteogenic markers [29]. The cell type may play an important role, since mouse cells are fast proliferating cells. Additionally, indirect effects of CAP on the formation of new bone after bioactivation of different surfaces have been described in the literature [61,62].

Nevertheless, it must be kept in mind that possible differences in the reaction of the cells may also be due to the different devices and forms of application of the plasma. For example, differences between our results and the study by Tominami et al. could be the plasma device, which is a helium-based plasma jet used to treat the cells for 5–10 s [29]. Further systematic studies with the same device on human osteoblasts are needed to clarify the effect of CAP on hard tissue cells.

In summary, in our study we were able to show effects of CAP on the proliferation and mineralization of HCO cells, which depended on the selected distance of the CAP device. Although the manufacturer recommends using the CAP device at a very close distance, this cannot always be guaranteed when using it on a patient who is breathing and moving. Thus, our data show that an application with some fluctuation in the distance to the cells still has an effect. However, it must be taken into account that a larger distance is potentially harmful to cells and tissues. Some plasma jets, for example, have been shown to generate mucosal side effects, such as erosion and inflammation at one day after CAP treatment, which, however, re-epitelize after 1 week [63]. Therefore, since the oral cavity is not a plane surface, but a three-dimensional space, further investigations are necessary to exclude whether these cytotoxic effects of CAP application are also detectable in the surroundings of the treated tissues, and whether they have any biological relevance.

Nevertheless, the use of CAP in dentistry offers an interesting possibility of faster healing of extraction sockets, as well as faster healing in implantations. In particular, people with impaired bone healing, such as patients receiving bisphosonates [64], might benefit from such therapy. Further in vitro and in vivo studies need to elucidate how the effect of CAP influences hard tissue healing in order to be routinely used in the future.

## 4. Materials and Methods

### 4.1. Study Design and Ethical Approval

This study evaluated effects of cold atmospheric plasma onto osteogenic differentiation potential in primary human osteoblast. Previously, cell viability was assessed based on cell vitality, proliferation, and cytotoxicity. Effects on osteogenic differentiation were assessed by gene expression analysis and protein expression analysis. The study was designed according to the Declaration of Helsinki and approved by the Ethics Committee of the Faculty of Medicine, University of Muenster (#2020-334-f-S).

### 4.2. Cold Atmospharic Plasma Conditions

A dielectric barrier discharge was used to create ambient air generated CAP at an output of 18 kV (Plasma ONE MEDICAL, Plasma MEDICAL SYSTEMS^®^ GmbH, Nassau, Germany) as previously described [23]. Various treatment conditions were used: Cells were treated at different treatment times (60–120 s) with the distance to the cell monolayer varying between 0.1 and 1.0 cm. The treatment was done 24 h after the adherence of the cells for the first time. This was repeated on the following days for the second and the third treatment, respectively.

### 4.3. Cell Culture

Primary human calvarial osteoblasts (HCO; 3H Biomedical, Uppsala, Sweden) were used. Conditions for cell culturing were described previously [65].

### 4.4. Cell Culture with CAP

Cells were seeded with a density of 10,000 cells/cm^2^ and allowed to adhere for 24 h. Various cell culture plates (Cellstar™, Greiner Bio One, Bad Nenndorf, Germany) were used. For the direct treatment, the culturing medium was removed immediately before the CAP application and was added freshly afterwards. For the indirect treatment, CAP was applied, leaving the cell culture medium on the cells. The distance indicated in the corresponding experiments was measured from the bottom of the dish and the CAP device was not in contact with the culturing medium. In the preliminary study part, various distances and impulse length were used to estimate nontoxic conditions. Osteogenic culturing medium was used [66,67]. In the main study part, effects onto osteogenic differentiation induced by CAP were analysed. Here, mineralization-inducing medium was used [66]. Cells were seeded with a density of 10,000 cells/cm^2^ and allowed to adhere for 24 h followed by a CAP treatment with two distances (0.1 and 0.5 cm). In addition, cells for the control group were cultivated without CAP treatment. Samples were analysed 4, 10, and 14 days after the CAP treatment. The cell culture part was repeated three times.

### 4.5. Cell Viability

The proliferation rate was estimated with an in-house MTT assay and cytotoxic effects were determined with the Pierce™ LDH Cytotoxicity Assay (Thermo Fisher Scientific, Wesel, Germany) described before [67]. The qualitative analysis of cell viability was performed via fluorescein diacetate/propidium iodide (FDA/PI) staining described previously [59].

### 4.6. RNA Extraction and Real-Time qPCR

RNA isolation, cDNA transcription, and real-time PCR analysis were described previously [23,67]. Specific commercially available primers (Qiagen, Hilden, Germany) were used. GAPDH was used as housekeeping gene. mRNA expression of *RUNX2, BMP2, ALP, TNFRSF11B, COL1A1, SPARC, SPP1, BGLAP, RANK, RANKL* was analysed.

### 4.7. Statistical Analysis

The experiments were performed in triplicates and repeated at least twice by calculating mean values and 95% confidence interval. For statistics, data were log2 transformed. The statistical analysis was performed using GraphPad Prism Software (GraphPad Software, San Diego, CA, USA). One-way ANOVA with post hoc Tukey’s multiple comparison test was used for *p* < 0.05.

## 5. Conclusions

CAP has a mediocre influence on differentiation potential of human osteoblasts. Cell viability is affected by the length of CAP treatment and by the distance of the CAP device to cells. Negative effects increased with longer durations of CAP treatment, with repetitions of CAP, and with larger distances to the cells. Nevertheless, CAP has positive effects on osteogenic differentiation potential. Regardless of shorter distances (0.1 or 0.5 cm), osteogenic marker genes are stimulated by CAP. Considering the bedside situation, these are negligible temporally delays in expression.

## Figures and Tables

**Figure 1 ijms-23-02503-f001:**
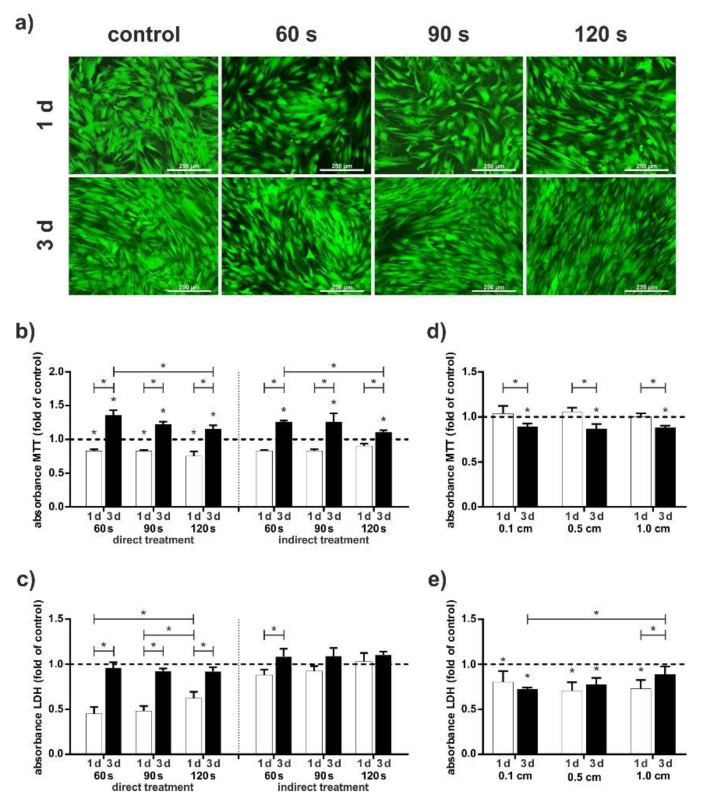
Influence of the different CAP treatment parameters on HCO cells as compared to untreated cells. (**a**–**c**) Distance of the CAP device to cells was constantly 2.0 cm; treatment length was 60, 90, and 120 s. Data at day 1 (white bar) and day 3 (black bar) are shown. (**d**,**e**) Distance of the CAP device to cells was 0.1, 0.5, and 1.0 cm with an indirect treatment. Data at day 1 (white bar) and day 3 (black bar) are shown. (**a**) Living/dead staining with a direct treatment. (**b**) Cell viability displayed by MTT assay with a direct (left site) and indirect (right site) treatment, n = 6. (**c**) LDH release after CAP treatment with a direct (left site) and indirect (right site) treatment, n = 6. (**d**) Cell viability displayed by MTT assay, n = 6. (**e**) LDH release after CAP treatment, n = 6. * Statistical significance (*p* < 0.05).

**Figure 2 ijms-23-02503-f002:**
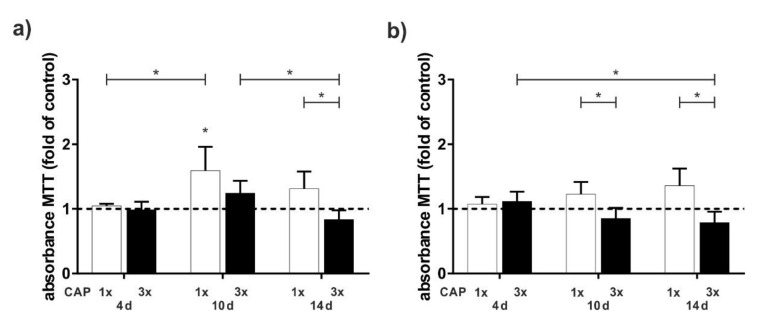
Influence of the different treatment times on HCO cells as compared to untreated cells. Cells were treated one time (white bars) and three times (black bars). (**a**) Cell viability at 0.1 cm distance displayed by MTT assay after 4, 10, and 14 days, n = 6. (**b**) Cell viability at 0.5 cm distance displayed by MTT assay after 4, 10, and 14 days, n = 6. * Statistical significance (*p* < 0.05).

**Figure 3 ijms-23-02503-f003:**
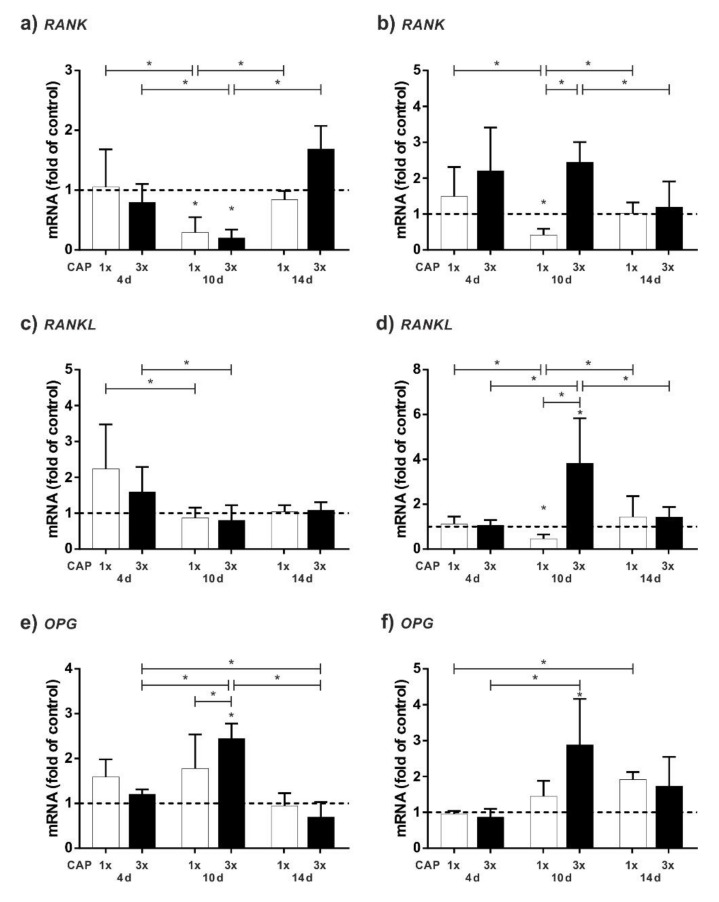
Influence of different CAP applications on HCO cells as compared to untreated cells. Cells were CAP-treated one time (white bars) and three times (black bars) and evaluated after 4, 10, and 14 days: (**a**) mRNA level of *RANK* at 0.1 cm distance; (**b**) mRNA level of *RANK* at 0.5 cm distance; (**c**) mRNA level of *RANKL* at 0.1 cm distance; (**d**) mRNA level of *RANKL* at 0.5 cm distance; (**e**) mRNA level of *OPG* at 0.1 cm distance; and (**f**) mRNA level of *OPG* at 0.5 cm distance. * Statistical significance (*p* < 0.05); n = 6.

**Figure 4 ijms-23-02503-f004:**
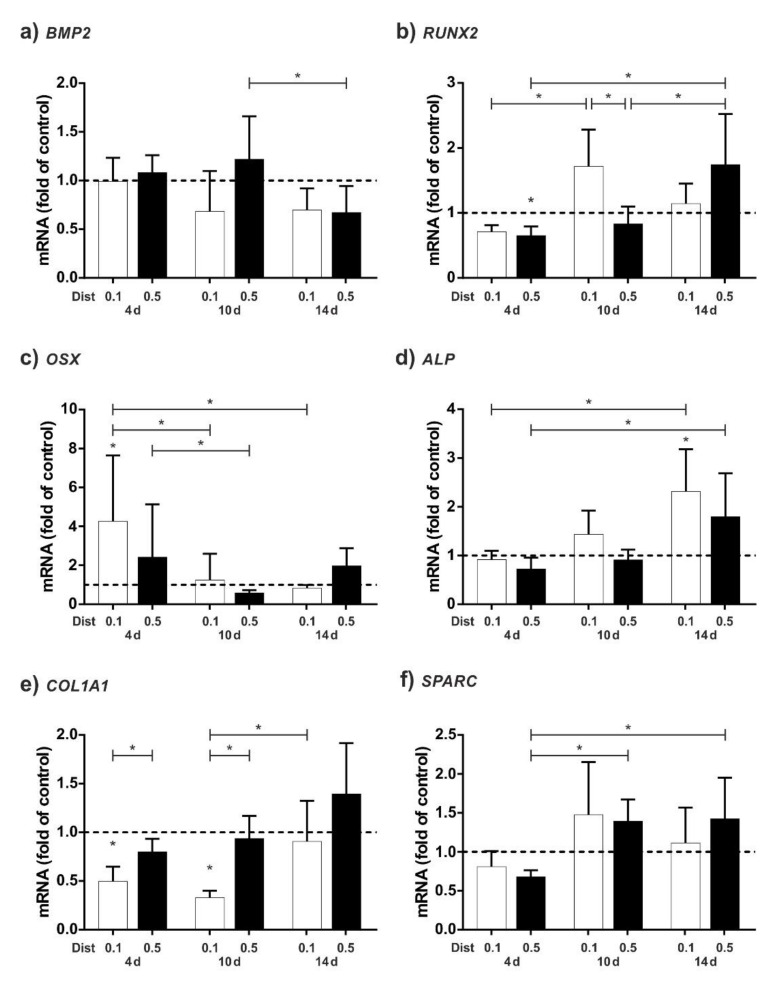
Influence of CAP on HCO cells as compared to untreated cells. Cells were CAP-treated at a distance of 0.1 cm (white bars) and 0.5 cm (black bars) and evaluated after 4, 10, and 14 days: (**a**) mRNA level of *BMP2*; (**b**) mRNA level of *RUNX2*; (**c**) mRNA level of *OSX*; (**d**) mRNA level of *ALP*; (**e**) mRNA level of *COL1A1*; and (**f**) mRNA level of *SPARC*. * Statistical significance (*p* < 0.05); n = 6.

## Data Availability

The datasets analysed during the current study are available from the corresponding author on reasonable request.

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
