# Peer review of "Effect of Cold Atmospheric Plasma (CAP) on Osteogenic Differentiation Potential of Human Osteoblasts"

_ijms, 2022, doi:10.3390/ijms23052503_

Round 1

Reviewer 1 Report

Overall the manuscript is sound. The authors do great work in describing the potential long-term proliferative effects of plasma treatment and the potential application of the treatment in a medical setting. There are clear next steps for the research that can build to a clinical trial.

However, the authors need to clarify their results and their comparisons to other plasma devices. Please see the detailed comments below:

Major comments:

  1. The phrase medium remained vs. medium changed is highly confusing. I originally took it as you replaced the same medium after treatment with no medium rather than changing to new medium after treatment with no medium. In the discussion, line 155, it is clarified to mean that the medium was left on during treatment, so your other electrode is the medium itself. This likely stymies the efficacy of plasma treatment for the DBD (volume of the medium is unknown since it was not added to the manuscript, or well-plate size, etc.). This makes sense as your treatment of the cells w/ medium shows no effect in comparison to the control, yet treatment directly on the cells shows less metabolic activity and less release of LDH 1 day after treatment compared to the control. These cells then recover after three days. This suggests that the direct treatment is negatively affecting the cells one day after treatment, and the cells recover after three days, exhibiting higher metabolic activity but not increased LDH secretion. The typical description in the plasma literature is to call treatment without media as direct treatment and treatment with media as an indirect treatment since plasma itself is a surface phenomenon and the DBD treats the surface of the liquid. This is stated later in the discussion but should be in Figure 1 and the description for it.

  1. Discussion hypothesizes that the higher distances showed cytotoxic effects by higher secretion of LDH, lines 137-140. For DBDs, the gap distance itself, especially larger ones > 2 mm, results in the formation of streamers. For adherent cell cultures, this can lead to voids in the monolayer post treatment that is either detachment of cells or cell death. Since streamers are localized energy in the plasma treatment, do you see these increase at large gap distances? The discussion here is a bit vague and the reactive species/UV/EM fields are usually more potent at smaller distances. Thus, seeing more cytotoxic effects at higher distances for DBDs is counterintuitive.

  1. Most of the discussion is just a description of the results while the results section is a brief summary of the results. I would consider changing it as the key findings laid out in the discussion are constantly interrupted by having to go back to the results section to analyze the graphs due to the minimal text describing the results that section.

  1. Throughout the manuscript, the author compares his results to literature, however, compares across plasma devices without carefully selecting direct DBD treatment in the literature, where it is an apt comparison. The literature describing plasma jets is a difficult comparison due to the very different physics and chemistry between that and a direct DBD treatment. Using CAP as a broad treatment for the sake of comparison is risky as it makes many assumptions between devices, most of which are incomparable even if they are of the same design (DBD vs. jet). I would try and limit this discussion to DBD treatment as the atmospheric plasma jet treatment is different as shown in the references cited beyond the author’s own studies. Are there any other studies that treat osteoblasts with a DBD with or without the presence of medium? Otherwise, the novelty of the finding is just for direct treatment and this comparison to other CAP devices is not all that important. It detracts from the main findings of the author’s study. If the author wants to include these references to compare across the literature, there needs to be a separate paragraph with a definitive statement that shows the author understands the differences between the treatments and discusses how these changes could lead to different cellular outcomes. It seems the author attempts this in the paragraph that begins on line 237, but that is after all the other comparisons are stated, which suggests that the earlier comparisons are apt when they are not.

Minor comments:

  1. Figure 1 caption and text is confusing. In the caption it says it only talks about the influence of 0.1, 0.5, and 1 cm. However, the text describes treatments at 2.0 cm. This is explicitly stated for graph a and is left for the reader to interpret it as also being described for b and c. There is no description in the figure caption. The 1d and 3d under the bars should be a larger text or the shading should be in a legend, it is very difficult to read.
  2. The treatments that show that the treatment height does not affect the proliferation of cells from 0.1 – 1 cm in c and d have no description as to whether the medium was changed after treatment between 1 day and 3 days.
  3. Line 97, Figure 2 caption, tree should be three.
  4. Figure 3 is not discussed in detail. I suggest adding text to the paragraph under Figure 2 that introduces the significance of Figure 3, starts with Figure 3a and discusses 3a and 3c since they are both completely absent from the results section.
  5. RANK, RANKL, and OPG are only briefly introduced in the abstract and not mentioned again till this paragraph. Please add the significance of studying these and what they represent for people who are interested in the topic but not experts in this area. I would repeat or move the information in the discussion (lines 170 – 173) to this part of the manuscript. I would do this as well for Figure 4.
  6. Line 142 - 148 – attain is usually not used in this way, I would consider rephrasing. Many components of CAP can travel 2 cm and affect cells. These lines say repeat the same outcome twice. Seeing as the effectors for the ‘positive’ and ‘negative’ effects of plasma treatment are not known at this time, it is premature to discuss which components can or cannot affect the cells.

Author Response

Overall the manuscript is sound. The authors do great work in describing the potential long-term proliferative effects of plasma treatment and the potential application of the treatment in a medical setting. There are clear next steps for the research that can build to a clinical trial.

Major comments:

However, the authors need to clarify their results and their comparisons to other plasma devices. Please see the detailed comments below:

  1. Concern of the Reviewer:

The phrase medium remained vs. medium changed is highly confusing. I originally took it as you replaced the same medium after treatment with no medium rather than changing to new medium after treatment with no medium. In the discussion, line 155, it is clarified to mean that the medium was left on during treatment, so your other electrode is the medium itself. This likely stymies the efficacy of plasma treatment for the DBD (volume of the medium is unknown since it was not added to the manuscript, or well-plate size, etc.). This makes sense as your treatment of the cells w/ medium shows no effect in comparison to the control, yet treatment directly on the cells shows less metabolic activity and less release of LDH 1 day after treatment compared to the control. These cells then recover after three days. This suggests that the direct treatment is negatively affecting the cells one day after treatment, and the cells recover after three days, exhibiting higher metabolic activity but not increased LDH secretion. The typical description in the plasma literature is to call treatment without media as direct treatment and treatment with media as an indirect treatment since plasma itself is a surface phenomenon and the DBD treats the surface of the liquid. This is stated later in the discussion but should be in Figure 1 and the description for it.

We thank the reviewer for the valuable comment. We see the reviewer's point and have changed the labelling to direct treatment vs. indirect treatment as suggested. However, it should be noted that indirect treatment in the plasma literature mainly describes treatments of fluids that are subsequently transferred to the biological target (Kaushik et al. 2018). In our case, the treatment of the medium took place above the cells, so that the biological effect was composed of an indirect and a (weaker) direct treatment. For this reason, we have additionally changed Material and Methods to clarify the definition of direct and indirect treatment:

L374ff:

For the direct treatment, the culturing medium was removed immediately before the CAP application and was added freshly afterwards. For the indirect treatment, CAP was applied, leaving the cell culture medium on the cells. The distance indicated in the corresponding experiments was measured from the bottom of the dish and the CAP device was not in contact with the culturing medium.

Kaushik NK, Ghimire B, Li Y, Adhikari M, Veerana M, Kaushik N, Jha N, Adhikari B, Lee SJ, Masur K, von Woedtke T, Weltmann KD, Choi EH. Biological and medical applications of plasma-activated media, water and solutions. Biol Chem. 2018 Dec 19;400(1):39-62. doi: 10.1515/hsz-2018-0226. PMID: 30044757.

Further, we additionally modified figure 1, changing the labelling to direct and indirect treatment and modified legend of figure 1(q.v. in manuscript).

Figure 1: Influence of the different CAP treatment parameters on HCO cells as compared to untreated cells. (a-c) Distance of the CAP device to cells was constantly 2.0 cm; treatment length was 60 s, 90 s, and 120 s. Data at day 1 (white bar) and day 3 (black bar) are shown. (d-e) Distance of the CAP device to cells was 0.1 cm, 0.5 cm, and 1.0 cm with an indirect treatment. Data at day 1 (white bar) and day 3 (black bar) are shown. (a) Living/dead staining with a direct treatment. (b) Cell viability displayed by MTT assay with a direct (left site) and indirect (right site) treatment, n=6. (c) LDH release after CAP treatment with a direct (left site) and indirect (right site) treatment, n=6. (d) Cell viability displayed by MTT assay, n=6. (e) LDH release after CAP treatment, respectively, n=6.

  1. Concern of the reviewer:

Discussion hypothesizes that the higher distances showed cytotoxic effects by higher secretion of LDH, lines 137-140. For DBDs, the gap distance itself, especially larger ones > 2 mm, results in the formation of streamers. For adherent cell cultures, this can lead to voids in the monolayer post treatment that is either detachment of cells or cell death. Since streamers are localized energy in the plasma treatment, do you see these increase at large gap distances? The discussion here is a bit vague and the reactive species/UV/EM fields are usually more potent at smaller distances.

We appreciate the reviewer’s comment. In preliminary experiments we have studied the effect of the DBD on the cells up to a distance of 2 cm above the surface of the medium. Interestingly, we did not see apoptotic effects using our device. However, since we wanted to use the device as suggested by the manufacturer, for the present study we have chosen closer distances to the cell monolayer. Nevertheless, the formation of streamers may be a possible explanation for the higher LDH secretion. We therefore thank the reviewer for his helpful comment and have also added this aspect to the discussion.

We have made the following changes to Discussion:

L182ff:

A distance-related increase in LDH secretion of cell monolayers has not been described for DBD plasma devices, which could be due to the fact that CAP has various components that have a different effect on cells, such as the production of reactive species or the emission of electromagnetic fields or ultraviolet radiation [35], which could have different effects at a greater distance. For example, other authors have described the effects of electromagnetic fields or UV light on LDH release [36,37]. It may be that at greater distances, only single components of the plasma affect the cells negatively. Furthermore, increased LDH secretion could also be caused by the formation of streamers, which have been observed at greater distances from DBDs [38,39]. Since the intensity of streamers is very high, cells may be negatively affected [40]. However, as the exact effects of CAP on cells and tissues are not yet fully understood, further research is needed to further elucidate the distance-related effects.

We have added the following reference:

[38] Pinchuk M, Nikiforov A, Snetov V, Chen Z, Leys C, Stepanova O. Role of charge accumulation in guided streamer evolution in helium DBD plasma jets. Sci Rep. 2021 Aug 26;11(1):17286. doi: 10.1038/s41598-021-96468-4. PMID: 34446766; PMCID: PMC8390516.

[39] Shcherbanev SA, Stepanyan SA, Popov NA, Starikovskaia SM. Dielectric barrier discharge for multi-point plasma-assisted ignition at high pressures. Philos Trans A Math Phys Eng Sci. 2015 Aug 13;373(2048):20140342. doi: 10.1098/rsta.2014.0342. PMID: 26170430.

[40] A. Y. Starikovskiy, N. L. Aleksandrov, and M. N. Shneider, J. Appl. Phys. 129, 063301 (2021). https://doi.org/10.1063/5.0037669

  1. Concern of the reviewer:

Most of the discussion is just a description of the results while the results section is a brief summary of the results. I would consider changing it as the key findings laid out in the discussion are constantly interrupted by having to go back to the results section to analyze the graphs due to the minimal text describing the results that section.

We would like to thank the reviewer for his/her extremely valuable comment, which helps us to improve the readability of the manuscript. We have followed the reviewer's suggestion and have modified the results. 

We have made the following changes to the text:

L102ff:

Next, we examined the effect of a single 60 s treatment versus multiple 60 s treatments distributed over several days, using a MTT assay. The CAP treatment resulted in an increase in cell viability, only after about 10 d, both at a distance of 0.1 cm and at a distance of 0.5 cm. There was a clear difference between single and multiple treatment, especially after 10 and 14d. Three treatments seem to have a negative effect on proliferation. Overall, cell proliferation was less pronounced at a greater distance between the CAP device and the cells (Fig. 2a, 2b)

 L117ff:

The focus of this study was the influence of CAP onto osteogenic differentiation potential. But, to ensure, that CAP has no effects on the contrary osteoclastic differentiation potential, expression of osteoclast-associates markers, RANK and RANKL, were analysed additionally.

Especially at a distance of 0.5 cm, multiple treatment causes an increase of RANK and RANKL (Fig. 3b, 3d). Especially after 10 days, the mRNA expression of both markers is strongly increased. Interestingly, the activation of RANK and RANKL is less pronounced at a closer distance of 0.1 cm. OPG, a decoy receptor of RANKL and an osteoclastogenic inhibition factor, shows the strongest expression after 10d (Fig.3e, 3f). In general, however, it appears that the upregulation of OPG mRNA expression is delayed with a 0.5 cm treatment compared to a 0.1 cm treatment (Fig. 3e, 3f). We observed an upregulation of OPG mRNA at both distances, 0.1 cm and 0.5 cm especially at 10 d. For 0.5 cm distance, OPG shows a similar expression pattern to RANKL. For a distance of 0.1 cm a different expression pattern to RANKL is visible. Only at day 4 similarities can be observed. At day 10, expression of OPG increased strongly in contrast to the expression of RANK and RANKL.

L141 ff:

For the following experiments, we focused on a single CAP treatment, as the triple application of CAP had more damaging effects on the cells Figure 4 summarised gene expression results of different early osteogenic markers and early mineralisation inducing marker. [Text interrupted by figure 4]

Interestingly, CAP had a slight influence on the regulation of BMP2 mRNA. However, early osteogenic markers, such as RUNX, and OSX are activated slightly faster by CAP at a distance of 0.1 cm than at a distance of 0.5 cm (Fig. 4a-c); however, this effect equalises over time and then the higher distance has a greater effect at 14 d. Overall, OSX regulation peaks at 4d, whereas RUNX 2 peaks at 10d for 0.1 cm distance and at 14d for 0.5 cm distance. Similarly, for mRNA regulation of ALP, the lower distance of the DBD causes a stronger effect than the higher distance, with the peak at 14d (Fig. 4d). However, for COL1A1, CAP results in an overall down-regulation of mRNA levels at 4 and 10d, with this being more pronounced at a distance of 0.1cm from the cells. Only at a distance of 0.5 cm at 14 d there is a slight upregulation of the COL1A1 mRNA level. Finally, CAP induces a slight upregulation of mRNA expression in SPARC, especially after 10 and 14d.

  1. Concern of the reviewer:

Throughout the manuscript, the author compares his results to literature, however, compares across plasma devices without carefully selecting direct DBD treatment in the literature, where it is an apt comparison. The literature describing plasma jets is a difficult comparison due to the very different physics and chemistry between that and a direct DBD treatment. Using CAP as a broad treatment for the sake of comparison is risky as it makes many assumptions between devices, most of which are incomparable even if they are of the same design (DBD vs. jet). I would try and limit this discussion to DBD treatment as the atmospheric plasma jet treatment is different as shown in the references cited beyond the author’s own studies. Are there any other studies that treat osteoblasts with a DBD with or without the presence of medium? Otherwise, the novelty of the finding is just for direct treatment and this comparison to other CAP devices is not all that important. It detracts from the main findings of the author’s study. If the author wants to include these references to compare across the literature, there needs to be a separate paragraph with a definitive statement that shows the author understands the differences between the treatments and discusses how these changes could lead to different cellular outcomes. It seems the author attempts this in the paragraph that begins on line 237, but that is after all the other comparisons are stated, which suggests that the earlier comparisons are apt when they are not.

We agree with the reviewer that a main focus should be laid on DBD devices. However, the application of DBDs to cell monolayers is limited in the literature, so discussion of all our results would not be possible. We agree that the lack of comparison to other researchers is a limitation of the study, but it highlights the need for further research in this area. We have therefore updated the literature wherever possible and have also compared our results to CAP effects of other devices. In these cases, we have clearly stated the devices used in the studies. We hope that the reviewer is comfortable with our changes.

At first, we added the following sentence to Introduction:

L 46:

Different technologies are known for medical applications, such as dielectric barrier discharge (DBD) devices, plasma jets or hybrid devices [21]

[21] Heinlin, J.; Morfill, G.; Landthaler, M.; Stolz, W.; Isbary, G.; Zimmermann, J.L.; Shimizu, T.; Karrer, S. Plasma Medicine: Possible Applications in Dermatology. JDDG: Journal der Deutschen Dermatologischen Gesellschaft 2010, 8, 968–976, doi:10.1111/j.1610-0387.2010.07495.x.

Additionally, we have made the following changes to Discussion:

L172ff:

At first, we wanted to find out the optimal parameter for CAP treatment, such as length of treatment or distance between the CAP device and the cells. Since our device is a DBD device, for which the manufacturer recommends a close distance to the tissue, which is not further specified, optimal parameters to osteoblast cell monolayer had to be investigated. Interestingly, there are few differences in viability, proliferation and cytotoxicity at a distance of 0.1 and 0.5 cm (Fig. 1d and 1e). Higher distances showed cytotoxic effects by higher secretion rate of LDH (Fig. 1c). Referring to the literature, Takahashi et al. have used an experimental DBD for irradiating human gingival fibroblast monolayers for 30 s at a distance of 1 cm and detected no difference in LDH release as compared to the control [33]. Steinbeck et al. observed a frequency-dependent increase in LDH release after 10s DBD treatment of osteoblasts at a distance of 2 mm [34]. Since a distance-related increase in LDH secretion of cell monolayers has not been described for DBD plasma devices, , which could be due to the fact that CAP has various components that have a different effect on cells, such as the production of reactive species or the emission of electromagnetic fields or ultraviolet radiation [35],…

 L203ff:

Additionally, CAP was found to have a proliferative effect on osteoblasts, as it has been shown in other studies for other cell types, such as gingival fibroblasts, MG63 cells or cementoblasts [23,32,33,41]. Interestingly, this effect was first observed after 10 days. In contrast, fibroblasts revealed a significant increase in proliferation already after 72h and cementoblasts and MG63 cells already after 24h after DBD treatment [32,33,41]. It appears that the proliferative effect in osteoblasts after stimulation with CAP only becomes apparent after a longer time than in the other cells. However, possible differences could also be due to the way CAP is applied: In our experiments, the cell culture medium was removed before applying CAP to show the direct effect on the cells, since in preliminary experiments, we observed negative results with remaining medium during CAP treatment (Fig. 1c). Steinbeck et al. have also removed culture media before the treatment and replaced it immediately after DBD treatment, but the authors did not examine cell proliferation but the number of dead cells [34]. The authors show a slight increase in dead cells due to CAP treatment at different intensity settings of the DBD, with the cells only being treated for 10 seconds. Since we did not observe any negative effects after 60 seconds, the difference could be due to the configuration of the plasma devices - Steinbeck et al used an experimental DBD and we used a certified medical device. However, further research is necessary to understand these effects in more detail. In many other studies found in the literature, the cell culture medium is left in place [32,41–43]. It is known that fluids, such as cell media can also be treated with CAP, both with plasma jets and DBDs, and have an attenuated biological effect on cells and tissues [44,45]. The overall CAP effect on cells covered with cell culture medium may be the result of simultaneous direct and indirect treatment. However, in our study, we wanted to focus on a direct treatment.

 [33] Takahashi, R.; Shimizu, K.; Numabe, Y. Effects of Microplasma Irradiation on Human Gingival Fibroblasts. Odontology 2015, 103, 194–202, doi:10.1007/s10266-014-0157-2.

[34] Steinbeck, M.J.; Chernets, N.; Zhang, J.; Kurpad, D.S.; Fridman, G.; Fridman, A.; Freeman, T.A. Skeletal Cell Differentiation Is Enhanced by Atmospheric Dielectric Barrier Discharge Plasma Treatment. PLoS One 2013, 8, e82143, doi:10.1371/journal.pone.0082143.

[42] Karki, S.B.; Yildirim-Ayan, E.; Eisenmann, K.M.; Ayan, H. Miniature Dielectric Barrier Discharge Nonthermal Plasma Induces Apoptosis in Lung Cancer Cells and Inhibits Cell Migration. Biomed Res Int 2017, 2017, 8058307, doi:10.1155/2017/8058307.

[43] Haertel, B.; Straßenburg, S.; Oehmigen, K.; Wende, K.; von Woedtke, T.; Lindequist, U. Differential Influence of Components Resulting from Atmospheric-Pressure Plasma on Integrin Expression of Human HaCaT Keratinocytes. Biomed Res Int 2013, 2013, 761451, doi:10.1155/2013/761451.

[45] Azzariti, A.; Iacobazzi, R.M.; Di Fonte, R.; Porcelli, L.; Gristina, R.; Favia, P.; Fracassi, F.; Trizio, I.; Silvestris, N.; Guida, G.; et al. Plasma-Activated Medium Triggers Cell Death and the Presentation of Immune Activating Danger Signals in Melanoma and Pancreatic Cancer Cells. Sci Rep 2019, 9, 4099, doi:10.1038/s41598-019-40637-z.

            L226ff:

Apoptotic effects are well known for longer treatments with CAP: In an in-vitro study, for example, treatment with DBDs longer than 40s results in significant decrease in cell number of HaCaT keratinocytes [43]. These apoptotic effects are especially observed for neoplastic cells using DBDs and plasma jet technology [26,42,43,46]. These selective effects on benign cells, such as fibroblasts, compared to cancer cells have also been described for indirect DBD treatment [47]. However, since the authors activated the media for 20 minutes and we only performed an indirect treatment of 30-90 seconds, this may explain the weak effect of the indirect treatment on the cell monolayers in the preliminary experiments.

However, we can conclude for our device that a direct treatment of 60 s is beneficial and multiple treatments are harmful for HCO. Further detailed studies are necessary to find out how the cells may react to treatment times with CAP between 60 s and 180 s.

[47] Han, I.; Choi, S.A.; Kim, S.I.; Choi, E.H.; Lee, Y.J.; Kim, Y. Improvement of Cell Growth of Uterosacral Ligament Fibroblast Derived from Pelvic Organ Prolapse Patients by Cold Atmospheric Plasma Treated Liquid. Cells 2021, 10, 2728, doi:10.3390/cells10102728.

L248f:

The authors show a reduction of RANKL and an increase of OPG in periodontitis in an animal model after a 2-minute treatment with an argon plasma jet [50]. However, in our experiments, we have not tested the effect of a 2-minute treatment with DBD CAP, but only that of a 3x1-minute treatment.

            L2701ff:

Further, we analyzed the expression of another osteogenic relevant marker: Surprisingly, BMP2 one of the potential inducers of osteogenic differentiation [11] has not been affected by CAP. Less is known about CAP and expression of BMP2. Studies focusing on the expression rate of osteogenic markers showed a relatively low expression level of BMP2 after both plasma jet and DBD CAP treatment [30,34]. In contrast, however, the reactive oxygen species that contribute to the CAP effect [35] have been described to control the expression of signaling molecules such as BMP2 to mediate osteoblast differentiation [54]. However, since the exact effect of CAP is not yet understood, it is possible that the sum of all components has a different effect than the isolated effect of ROS/RNS, electric fields or UV light. For this reason, additional research is needed to further understand the CAP effect on osteogenic differentiation.

L291f:

These differences may be caused by the fact that murine cementoblasts were used and treated indirectly (cells with remaining culturing medium). Further studies on murine osteoblasts or human cementoblasts could help to decipher these questions.

L313ff:

Regarding mineralization, expression rate of ALP, which is also a key factor of bone mineralization [60], was upregulated by CAP. Other authors have also investigated the effect of CAP on the mineralization of hard tissues: Interestingly, CAP treatment of murine pre-osteoblasts covered with cell-culture medium using a plasma jet resulted in higher proliferation, ALP activity, and upregulation of osteogenic markers [29]. The cell type may play an important role, since mouse cells are fast proliferating cells. Additionally, indirect effects of CAP on the formation of new bone after bioactivation of different surfaces have been described in the literature [61,62].

 [61] Moriguchi, Y.; Lee, D.-S.; Chijimatsu, R.; Thamina, K.; Masuda, K.; Itsuki, D.; Yoshikawa, H.; Hamaguchi, S.; Myoui, A. Impact of Non-Thermal Plasma Surface Modification on Porous Calcium Hydroxyapatite Ceramics for Bone Regeneration. PLoS One 2018, 13, e0194303, doi:10.1371/journal.pone.0194303.

[62] Zuo, J.; Huang, X.; Zhong, X.; Zhu, B.; Sun, Q.; Jin, C.; Quan, H.; Tang, Z.; Chen, W. A Comparative Study of the Influence of Three Pure Titanium Plates with Different Micro- and Nanotopographic Surfaces on Preosteoblast Behaviors. J Biomed Mater Res A 2013, 101, 3278–3284, doi:10.1002/jbm.a.34612.

L333f:

Some plasma jets, for example, have been shown to generate mucosal side effects, such as erosion and inflammation at one day after CAP treatment, which, however, reepitelize after 1 week [62].

  1. Concern of the reviewer: 

Minor comments: Figure 1 caption and text is confusing. In the caption it says it only talks about the influence of 0.1, 0.5, and 1 cm. However, the text describes treatments at 2.0 cm. This is explicitly stated for graph a and is left for the reader to interpret it as also being described for b and c. There is no description in the figure caption. The 1d and 3d under the bars should be a larger text or the shading should be in a legend, it is very difficult to read.

The treatments that show that the treatment height does not affect the proliferation of cells from 0.1 – 1 cm in c and d have no description as to whether the medium was changed after treatment between 1 day and 3 days.

Line 97, Figure 2 caption, tree should be three.

We thank the reviewer for his/her valuable comment. We fully agree with the reviewer and apologise for the inaccuracy. We have adapted the legend of the figure and changed the figure itself. Furthermore, we have revised all the figures (q.v. in manuscript) and figure legends carefully for a better comprehensibility:

Figure 1: Influence of the different CAP treatment parameters on HCO cells as compared to untreated cells. (a-c) Distance of the CAP device to cells was constantly 2.0 cm; treatment length was 60 s, 90 s, and 120 s. Data at day 1 (white bar) and day 3 (black bar) are shown. (d-e) Distance of the CAP device to cells was 0.1 cm, 0.5 cm, and 1.0 cm with an indirect treatment. Data at day 1 (white bar) and day 3 (black bar) are shown. (a) Living/dead staining with a direct treatment. (b) Cell viability displayed by MTT assay with a direct (left site) and indirect (right site) treatment, n=6. (c) LDH release after CAP treatment with a direct (left site) and indirect (right site) treatment, n=6. (d) Cell viability displayed by MTT assay, n=6. (e) LDH release after CAP treatment, respectively, n=6.

Figure 2. Influence of the different treatment times on HCO cells as compared to untreated cells. Cells were treated one time (white bars) and three times (black bars), respectively. (a) Cell viability at 0.1 cm distance displayed by MTT assay after 4 days, 10 days, and 14 days, respectively, n=6. (b) Cell viability at 0.5 cm distance displayed by MTT assay after 4 days, 10 days, and 14 days, respectively, n=6. * statistical significance (p<0.05).

Figure 3. Influence of different CAP applications on HCO cells as compared to untreated cells. Cells were CAP-treated one time (white bars) and three times (black bars) and evaluated after 4 days, 10 days, and 14 days, respectively. (a) mRNA level of RANK at 0.1 cm distance. (b) mRNA level of RANK at 0.5 cm distance. (c) mRNA level of RANKL at 0.1 cm distance. (d) mRNA level of RANKL at 0.5 cm distance. (e) mRNA level of OPG at 0.1cm distance. (f) mRNA level of OPG at 0.5 cm distance. n=6. * statistical significance (p<0.05).

Figure 4. Influence of CAP on HCO cells as compared to untreated cells. Cells were CAP-treated at a distance of 0.1 cm (white bars) and 0.5 cm (black bars) and evaluated after 4 days, 10 days, and 14 days, respectively. (a) mRNA level of BMP2. (b) mRNA level of RUNX2. (c) mRNA level of OSX. (d) mRNA level of ALP. (e) mRNA level of COL1A1. (f) mRNA level of SPARC.  n=6. * statistical significance (p<0.05).

  1. Concern of the reviewer:

Figure 3 is not discussed in detail. I suggest adding text to the paragraph under Figure 2 that introduces the significance of Figure 3, starts with Figure 3a and discusses 3a and 3c since they are both completely absent from the results section. RANK, RANKL, and OPG are only briefly introduced in the abstract and not mentioned again till this paragraph. Please add the significance of studying these and what they represent for people who are interested in the topic but not experts in this area. I would repeat or move the information in the discussion (lines 170 – 173) to this part of the manuscript. I would do this as well for Figure 4.

We thank the reviewer for his very helpful comment. We see the reviewers point and have therefore updated the results. Figure 3 shows the results of the osteoclastic-related genes RANK and RANKL, and it decoy receptor osteoprotegerin (OPG). It is well known, that CAP has negative effects as well. This type of study usually analyses the expression of osteogenesis-related markers. Possible opposite effects are not being studied in many cases. OPG is not only the decoy receptor of RANKL and an osteoclastic inhibition factor, but also an important osteogenic related marker. We chose to combine gene expression results of RANK/RANKL and OPG together in figure 3 to point out their contrary expression rate.

To clarify the importance and the reason of study also osteoclast-associated marker, we added the following sentences to the results and to discussion.

L117f:

The focus of this study was the influence of CAP onto osteogenic differentiation potential. But, to ensure, that CAP has no effects on the contrary osteoclastic differentiation potential, expression of osteoclast-associates markers, RANK and RANKL, were analysed additionally.

L124f:

OPG, which is a decoy receptor of RANKL and an osteoclastogenic inhibition factor, shows the strongest expression after 10d (Fig.3e, 3f)

L239ff:

We also analysed expression of osteoclast-associated markers. Usually, in studies investigating osteoblast expression patterns, mainly alteration in expression of well-known osteogenic markers is analysed. Possible opposite effects are untended.

L256ff:

Due to the facts that OPG is a decoy receptor for RANKL, we could observe a nearly similar expression pattern of OPG and RANKL. This could be interpreted that OPG estruses RANKL. For 0.1 cm distance, expression pattern of OPG and RANKL are similar at day 4 and at day 10 expression of OPG increased strongly. OPG does not only inhibits osteoclastogenesis it is also an early marker for osteogenic differentiation [53].

            L270f:

Further, we analysed the expression of another main osteogenic relevant marker with regard to early marker and early mineralisation inducing markers.

  1. Concern of the reviewer:

Line 142 - 148 – attain is usually not used in this way, I would consider rephrasing. Many components of CAP can travel 2 cm and affect cells. These lines say repeat the same outcome twice. Seeing as the effectors for the ‘positive’ and ‘negative’ effects of plasma treatment are not known at this time, it is premature to discuss which components can or cannot affect the cells.

We thank the reviewer for the valuable comment. As suggested, we have rephrased the entire paragraph. We have made the following changes to the text, as already mentioned above.

Reviewer 2 Report

The study by Eggers et al. investigated the effect of cold atmospheric plasma on osteoblasts cultures. Although the topic is interesting and deserves attention due to the clinical application, some conclusions were not supported by the data, as follow:

  • The authors state that “Overall, however, distances of 1.0 caused the strongest negative effects on the cells”. However, the data presented in the Fig 1 do not indicate any significant difference on the cells proliferation with the distance of CAP application.
  • It is not clear the reason of investigating the osteoclast-associated markers RANK and RANKL in osteoblasts cultures.
  • the most important points: Additionally, in the experimental section the authors state “In the main study part, effects onto osteogenic differentiation and mineralization capability induced by CAP were analyzed”. (page 10, line 289) Actually, the mineralization capacity was not investigated in the study, since no data about the formation of mineral nodules (i.e. Alizarin red staining, SEM images, Ca/P mineral molar ratio) nor quantification of the collagen-I expression was presented. Moreover, since the cells were already mature osteoblasts cultivated on DMEM to induce differentiation and osteogenic medium, previously to CAP application, it is difficult to claim that the focus of the study was on the effect of CAP on the cells differentiation.

Author Response

  1. Concern of the reviewer:

The authors state that “Overall, however, distances of 1.0 caused the strongest negative effects on the cells”. However, the data presented in the Fig 1 do not indicate any significant difference on the cell’s proliferation with the distance of CAP application.

Thank you for your suggestion. We have to concede. Study design is limiting. Aware of it, we consider it in our statements. 

The observed effects are not so obviously. As we wrote a few sentences before, that the proliferation was not affected by the chosen distance. But we could observe a higher release of LDH to culturing medium and, so, a level of cytotoxicity. Summing up gained data, we conclude the distance of 1.0 cm showed the most negative effects comparing to distance 0.1 cm and 0.5 cm.   

L82:

To summarize, distances…

  1. Concern of the reviewer:

It is not clear the reason of investigating the osteoclast-associated markers RANK and RANKL in osteoblasts cultures.

The reviewer is right. To investigate expression of osteoclast-associated marker was not part of our study design firstly. It was more a stroke of luck to investigate them. It is well known, that CAP has negative effects as well. Usually in this kind of studies, expression of osteogeneses related genes are analyzed. Possible opposite effects are untended. We discussed this, we had the primer, the cDNA, and the man power, so why not analyzing them. And we observed significant effects in expression of osteoclast-associated markers when CAP treatment is repeated daily.

To clarify the importance of study also osteoclast-associated marker, we added following sentences in results and in discussion.

L117ff:

The focus of this study was the influence of CAP onto osteogenic differentiation po-tential. But, to ensure, that CAP has no effects on the contrary osteoclastic differentiation potential, expression of osteoclast-associates markers, RANK and RANKL, were analysed additionally.

L239ff:

We also analysed expression of osteoclast-associated markers. Usually, in studies investigating osteoblast expression patterns, mainly alteration in expression of well-known osteogenic markers is analysed. Possible opposite effects are untended.

  1. Concern of the reviewer:

The most important points: Additionally, in the experimental section the authors state “In the main study part, effects onto osteogenic differentiation and mineralization capability induced by CAP were analyzed”. (page 10, line 289) Actually, the mineralization capacity was not investigated in the study, since no data about the formation of mineral nodules (i.e. Alizarin red staining, SEM images, Ca/P mineral molar ratio) nor quantification of the collagen-I expression was presented. Moreover, since the cells were already mature osteoblasts cultivated on DMEM to induce differentiation and osteogenic medium, previously to CAP application, it is difficult to claim that the focus of the study was on the effect of CAP on the cells differentiation.

Thank you for your remainder. We did analyse changes in mineralization capability. We showed in figure 4 gene expression results for collagen 1 alpha 1(COL1A1), of osteonectin (SPARC), of osterix (OXN) and of ALP.  Further, we analysed alteration in mineralization rate also on protein level. We quantified mineralization rate with an “Alizarin Red S Staining Quantification Assay” (ScienCell #8678). We could not observe a significant earlier start of mineralization induced by CAP. In ELISA, detection of osteocalcin was not possible. Nonetheless, amount of osteocalcin was below sensitivity of used ELISA. Nor was the expression on gene level clearer and convincing. The results were not statistically significant and not comprehensible. We assume that this observed irritating results based on the strong limitation of the study by experimental design and has to be reanalysed in further studies. Summing up all gained results, we decided to analysed only earlier marker in osteogenesis and mineralization. The giving results in this manuscript should be used as helpful hints for further studies in this field of research.

We deleted in material and methods the phrase “and mineralization capability”.

                L383ff:

Osteogenic culturing medium was used [66,67]. In the main study part, effects onto osteogenic differentiation and mineralization capability induced by CAP were analysed.

L256ff:

Due to the facts that OPG is a decoy receptor for RANKL, we could observe a nearly similar expression pattern of OPG and RANKL. This could be interpreted that OPG estruses RANKL. For 0.1 cm distance, expression pattern of OPG and RANKL are similar at day 4 and at day 10 expression of OPG increased strongly. OPG does not only inhibits osteoclastogenesis it is also an early marker for osteogenic differentiation [53].

Round 2

Reviewer 2 Report

The authors have addressed all the questions raised by the reviewer.